# Detection of Bioactive Peptides’ Signature in Podolica Cow’s Milk

**DOI:** 10.3390/foods14050877

**Published:** 2025-03-04

**Authors:** Rosario De Fazio, Antonella Di Francesco, Pierluigi Aldo Di Ciccio, Vincenzo Cunsolo, Domenico Britti, Carmine Lomagistro, Paola Roncada, Cristian Piras

**Affiliations:** 1Department of Health Sciences, Magna Græcia University of Catanzaro, 88100 Catanzaro, Italy; rosario.defazio@studenti.unicz.it (R.D.F.); britti@unicz.it (D.B.); roncada@unicz.it (P.R.); 2Laboratory of Organic Mass Spectrometry, Department of Chemical Sciences, University of Catania, 95124 Catania, Italy; antonella.difrancesco@unict.it (A.D.F.); vcunsolo@unict.it (V.C.); 3Department of Veterinary Sciences, University of Turin, Largo Braccini 2, Grugliasco, 10095 Torino, Italy; pierluigialdo.diciccio@unito.it; 4Interdepartmental Center Veterinary Service for Human and Animal Health, University “Magna Graecia” of Catanzaro (CISVetSUA), 88100 Catanzaro, Italy; 5Associazione Regionale Allevatori Calabria (A.R.A. Calabria), 88046 Lamezia Terme, Italy; lomagistro.c@aracalabria.it

**Keywords:** Podolica milk, bottom-up proteomics, top-down peptidomics, bioactive peptides

## Abstract

The aim of this study was to identify and characterize the bioactive peptide profile of Podolica cow’s milk. This dairy product is known for its nutritional properties related to the presence of peculiar lipids and is a typical breed traditionally reared in southern Italy. Using top-down peptidomics, we identified 2213 peptides in milk samples from four different farms, with 19 matching bioactive sequences. Bioactivities include dipeptidyl peptidase-IV (DPP-IV) inhibition, angiotensin-converting enzyme (ACE) inhibition, antioxidant activity, enhanced calcium uptake, and other peptides with potential antimicrobial effects. DPP-IV-inhibitory peptides (e.g., LDQWLCEKL and VGINYWLAHK) suggest potential for type 2 diabetes management, while ACE inhibitors (such as YLGY and FFVAPFPEVFGK) could support cardiovascular health by reducing hypertension. Antimicrobial peptides such as SDIPNPIGSENSEK and VLNENLLR showed broad spectrum of activity against various harmful microorganisms, positioning Podolica milk as a promising source for natural antimicrobial agents. Additionally, peptides with osteoanabolic, antianxiety, and immunomodulatory properties further highlight the multifaceted health benefits associated with this type of milk. Our findings underline the functional richness of Podolica milk peptides with various bioactivity properties, which could enhance the value of derived dairy products and contribute to sustainable agricultural practices. Future research will aim to explore these bioactivity properties in vivo, establishing a foundation for functional foods and supplements based on Podolica milk.

## 1. Introduction

The Podolica cattle breed is native to southern Italy [1]. Podolica cows are known for their low milk yields with an exceptional milk quality in terms of nutritional properties, as well as for producing highly valued meat. Most of the milk is used to make dairy products (for ex. “Caciocavallo Podolico” cheese), which is made exclusively from Podolica milk [2]. Ensuring the authenticity of this product requires affordable and easy-to-implement methods for detecting pure Podolica milk or identifying mixtures with milk from other breeds [3]. Adulteration of high-value milk with that of lower-value breeds or species is common, which can compromise both the quality and authenticity of the milk, potentially leading to allergic reactions in consumers and reducing the amount of bioactive molecules with beneficial effects [4,5].

Two different methods for the differentiation of Podolica cow’s milk have been already described. The first one is based on Fourier Transform Infrared Spectroscopy [6] coupled with machine learning. The results showed that FTIR spectral profiling combined with machine learning effectively identified non-Podolica milk. Furthermore, by controlling for environmental variables, the method improved in accuracy, allowing for the detection of even smaller amounts of non-Podolica milk as a contaminant [6]. The second one is based on another rapid spectral profiling approach using LAP-MALDI and machine learning [7]. In this study, LAP-MALDI MS combined with linear discriminant analysis (LDA) machine learning was used as a rapid classification method to distinguish Podolica milk from that of other bovine breeds.

The results demonstrated that the MS profiles could effectively differentiate between Podolica and control milk samples. The classification accuracy was 86% in the training set and reached 98.4% in the test set [7].

On top of this, both differential protein representation and microorganism consortia were evaluated. Bottom-up proteomics combined with bacterial and somatic cell enrichment provided valuable insights into milk composition, identifying 220 proteins using a Bos taurus database. Cathelicidins and annexins were found at higher levels in control cows’ milk. Cathelicidins, antimicrobial peptides involved in innate immunity, and annexins, known for their role in inflammation and tissue repair, suggest an enhanced immune response in control cows, likely influenced by genetic or environmental factors [7].

Further analysis of microbial composition revealed significant differences between milk samples. Podolica milk exhibited lower levels of Firmicutes and lacked lactobacilli, a which were present in control samples. Clostridia, more abundant in Podolica milk, may play a probiotic and anti-inflammatory role [7].

Bovine milk is a rich source of bioactive compounds as lysozyme, lactoferrin, immunoglobulins, growth factors, and hormones. Additionally, bioactive peptides can be released through the enzymatic breakdown of milk proteins [8]. Typically, these peptides remain latent within the primary structure of milk proteins until they are liberated via proteolysis, primarily from casein fractions (α-, β-, γ-, and κ-casein) and whey proteins (such as β-lactoglobulin, α-lactalbumin, serum albumin, immunoglobulins, lactoferrin, and protease-peptone fractions). The growing body of evidence highlighting the health benefits of these milk-derived bioactive peptides has positioned them as promising candidates for inclusion in functional foods [9,10].

Peptide generation occurs through enzymatic hydrolysis or microbial fermentation, either within the body during digestion—where digestive enzymes like trypsin and gut microbiota enzymes play a role—or externally, during food processing, ripening, or through in vitro enzymatic hydrolysis. Once absorbed into the bloodstream from the intestines, bioactive peptides can exert both localized effects within the gastrointestinal system and systemic physiological benefits. Notably, microbial and digestive enzymes target different cleavage sites within protein sequences, leading to variations in the peptides produced. For instance, the peptide KVLPVP, derived from casein, exhibits significantly enhanced angiotensin-converting enzyme (ACE) inhibitory activity following additional pancreatic digestion compared to the peptide KVLPVPQ, which is generated solely through proteolysis by *Lactobacillus helveticus* [11].

From methodological perspective, proteomics and peptidomics represent essential tools for discovering bioactive peptide profiles in bovine milk. Proteomics identifies milk proteins and their modifications, while peptidomics focuses on naturally occurring peptides generated by enzymatic hydrolysis. These approaches help characterize peptide sequences, their bioactivities, and their roles in nutrition and health. Advanced methodologies, including fractionation, mass spectrometry, and bioinformatics, enhance peptide identification and functional analysis. By integrating multi-omics strategies (proteomics, peptidomics, and metabolomics), researchers can track peptide release, assess their bioavailability, and explore their potential benefits, such as DPP-IV inhibition for metabolic health applications [12].

Considering this premise, it may be hypothesized that a specific bioactive peptide signature could be representative of Podolica milk. In regard to this, the aim of this work was to identify and characterize the bioactive peptides in Podolica milk, with a focus on peptides linked to its nutritional and health-promoting properties. By profiling these peptides, we seek to better understand their potential roles in immune modulation, antimicrobial activity, and overall contribution to the milk’s quality.

## 2. Materials and Methods

### 2.1. Sampling Plan

The Podolica herds consisted of 50, 35, 40, and 30 cows (total 155), all grazing on Mediterranean pastures with no supplementary feed, except for polyphyte hay during the coldest winters. In the summer season, the cows were moved to mountain pastures at 1500 m above sea level.

Cows that had received antibiotic treatments in the previous two months were excluded from sampling plan to minimize the influence of the antibiotic use.

The study focused on bulk milk samples collected from four Podolica farms but, when it was necessary to compare the datasets with other than Podolica milk, another four farms were included (control) with cows from Frisona, Pezzata Rossa, and Bruna Alpina breeds (one farm had mixed Frisona and Pezzata Rossa).

For each farm, two sampling sessions at two different timepoints were performed, at least a week apart to capture variability over time. For each sampling session, two different samples were collected from bulk tank with the temperature stable at 2 °C from at least 1 h and after at least 10 min under gentle stirring. The sample collection was performed from the top with a clean dipper; the collection from the bulk tank after the milking sessions and after the temperature is stable allows to have a sample representing the mean of the entire farm. Considering that 4 different Podolica farms were sampled in two different sessions and that, for each session two different samples were collected and separately processed, in total, 4 samples were obtained from each farm and separately analyzed.

### 2.2. Sample Preparation

Protein concentrations were measured using the Qubit Protein Assay Kit with a Qubit 1.0 Fluorometer (ThermoFisher Scientific, Milan, Italy) [13]. Approximately 50 µg of protein from each extract was reduced with 39 µg of DTT (for 3 h at 20 °C), followed by alkylation with 94 µg of IAA (in the dark for 1 h at 20 °C). Samples were digested overnight with porcine trypsin at a 1:50 enzyme/substrate ratio (37 °C). A 5% aqueous formic acid solution was added to bring the final concentration to 25 ng/µL, with a total volume of 2 mL.

### 2.3. LC-MS MS Analysis

MS/MS data were acquired on a ThermoFisher Scientific Orbitrap Fusion Tribrid^®^ mass spectrometer (ThermoFisher Scientific, Bremen, Germany) [13]. Liquid chromatography was performed using a Dionex UltiMate 3000 RSLC nano system (Thermo Scientific; Sunnyvale, CA, USA). One microliter of peptide mixture was loaded onto an Acclaim^®^ Nano Trap C18 Column (100 μm i.d. × 2 cm, 5 μm particle size, 100 Å pore size), followed by a wash with solvent A (0.1% FA in H_2_O) for 3 min at 7 μL/min. Peptides were eluted onto a PepMap^®^ RSLC C18 EASY-Spray column (Thermo Scientific; Sunnyvale, CA, USA; 75 μm i.d. × 50 cm, 2 μm particle size, 100 Å pore size) and separated at 40 °C using a 0.25 μL/min flow rate with a gradient, as previously reported [14].

Peptides were ionized using electrospray ionization (ESI) at 1.75 kV and introduced via a heated ion transfer tube (275 °C). Full-scan MS was conducted from *m*/*z* 200 to 1600 at 120,000 resolution (at *m*/*z* 200), with MS/MS performed on precursor ions isolated at 1.6 Th using the quadrupole. HCD fragmentation (35 eV) and low-resolution MS/MS analysis were carried out in the linear ion trap. Precursors with 2–4 charges and an intensity above 5 × 10^3^ counts were selected for MS/MS. Dynamic exclusion was set to 60 s with a 10 ppm tolerance for precursor ions and isotopes. Monoisotopic precursor selection was enabled. The mass spectrometer operated in top-speed mode, cycling through MS/MS for a maximum of 3 s. Spectral quality was enhanced using parallelizable time, maximizing MS/MS precursor ion injection. The instrument was calibrated using the Pierce^®^ LTQ Velos ESI Positive Ion Calibration Solution (ThermoFisher Scientific). Triplicate RP-nHPLC/nESI-MS/MS analyses were performed for reproducibility.

### 2.4. Bioinformatics Analysis

Label-free quantification (LFQ) of the MS/MS data was carried out using MaxQuant software (version 2.4.2.0) [15]. Peak lists were searched against the UniProt *Bos taurus* database (as previously mentioned), a bacteria database (filtered by taxonomy, reviewed, 336,171 entries, 3 July 2023), and the CARD database (5010 reference sequences, January 2023) [16]. For these searches, the precursor mass tolerance was set to 20 ppm for the initial search and 4.5 ppm for the main search, with a fragment mass tolerance of 0.5 Da. The searches were performed with the parameter “no digestion” and “unspecific digestion”. Variable modifications included methionine oxidation and N-terminal acetylation, while carbamidomethylation of cysteines was set as a fixed modification. A 1% false discovery rate (FDR) was applied at the protein level. The LFQ minimum number of neighbors was set to 3, with an average of 6.

Peptide data from the bacterial database search were analyzed using the UNIPEPT web tool (https://unipept.ugent.be/mpa, accessed on 29 July 2024) for the GO analysis [17].

The list of peptides yielded by MaxQuant was then searched for the presence of bioactive ones in the “Milk Bioactive Peptide Database” (https://mbpdb.nws.oregonstate.edu/, last accessed on 30 January 2025) [18,19].

## 3. Results

The overall analysis aimed at the detection of peptides with biological activities present in Podolica milk. As described in the methods section, the analysis was performed on eight different Podolica’s milk samples from four farms (bulk tank) and on eight other different samples from four farms of different breeds (see Section 2). For completeness, the overall composition of abundant proteins is shown in Appendix A. Briefly, according to this analytical method, the composition of high abundant proteins of Podolica and control milk is essentially the same with no substantial differences.

Top-down analysis performed using the *Bos taurus* database (UNIPROT, taxonomy, reviewed, date of download, specify in methods as well) and the function “no digestion” search through the MaxQuant software yielded the results presented in Table 1. The first identified protein is Secretin which is a hormone involved in different processes, such as regulation of the pH of the duodenal content, food intake, and water homeostasis. The second identified protein is annotated as Protein Hook homolog 3 and, similar to all the reviewed UNIPROT proteins, its existence was demonstrated at protein level.

Considering the possible activity of the numerous proteases present in milk, a peptides search was performed using the same database, but considering the possibility of detecting peptides that may have originated by several cleavage processes. This was achieved using the MaxQuant function cleavage “unspecific” that takes into consideration the most probable proteases intervention. In total, 2213 peptides were identified in Podolica milk (2092 in control milk) that were found in the Milk Bioactive Peptide Database (https://mbpdb.nws.oregonstate.edu/, accessed on 29 July 2024) [18,19]. Among those, the search yielded 19 hits in total (Table 2), all belonging from sequences of *Bos taurus* originating from 14 peptides, as shown in the Venn diagram in Figure 1. A total of 10 peptides were specifically present and unique to Podolica milk. On the other hand, five peptides from the control cows matched some bioactivity, four commonly shared with Podolica milk and one only found in control milk (VRYL) with reported ACE-inhibitory function. The four commonly shared peptides were recorded with the following functions: LDQWLCEKL- DPP-IV-inhibitory; VGINYWLAHK- ACE-inhibitory, antioxidant, DPP-IV-inhibitory; HPHPHLSF-ACE-inhibitory, osteoanabolic; YIPIQYVLSR-immunomodulatory, opioid. The pie chart in Figure 2 represents a more comprehensive view about the functions of all the bioactive peptides detected in Podolica milk. Two of the identified peptides have dipeptidyl peptidase 4 inhibition function, six have ACE-inhibitory function, two are antioxidant, two are responsible for increasing calcium uptake, three have antimicrobial activity, and the remaining ones have antianxiety, osteoanabolic, immunomodulatory, and analgesic (opioid-like) functions. All the identified peptides belong to four different precursor proteins, specifically, alpha-lactalbumin, alpha-S1-casein, and kappa-casein.

As in Figure 3, UNIPEPT GO analysis about the functions of the detected peptides highlighted immune response, antimicrobial humoral response, and the defense against Gram + bacteria as the key biological processes.

*DPP-IV-inhibitory peptides,* as in Table 3 (green), originate from two diverse parts of bovine alpha-lactalbumin, and the “VGINYWLAHK” has an IC50 of 13.8 µM.

*ACE-inhibitory peptides* (Table 4, blue) were a major part of the bioactive peptides detected; among those, one originated from alpha-lactalbumin, four from alpha-S1-casein, and one from kappa-casein. This latter’s bioactivity was originally annotated when it was isolated from sheep’s milk; however, its sequence is present in *Bos taurus* as well. All the functions of these detected peptides will be better described in the Section 4.

*Antioxidant peptides* were annotated as being involved in functions and belonged from two different proteins (alpha-lactalbumin and alpha-S1-casein, see Table 5, yellow) and other two, both originating from alpha-S1-casein, are important to enhance calcium uptake (Table 6, grey).

*Antimicrobial peptides* (three different) all originated from alpha-S1-casein and are responsible for antimicrobial function against diverse bacteria as *L. innocua*, *E. coli*, *C. sakazakii*, and *B. subtilis* (Table 7, pink).

Other peptides with miscellaneous functions are summarized in Table 8 (white); four different detected peptides have functions including antianxiety, osteoanabolic, immunomodulatory, and opioid. One originated from alpha-S1-casein, and two from kappa-casein. One of these latter was demonstrated to be present in sheep.

As in the following Figure 4, the peptide VLNENLLR were found to be more represented in Podolica cow’s milk. The difference was found to be statistically significant when applying the Wilcoxon test (non-parametric) with *p*-values of 0.0348.

## 4. Discussion

This study focused on identifying and characterizing bioactive peptides in Podolica milk, using advanced proteomic techniques to enhance our understanding of its nutritional and health-promoting properties. The identification of significant proteins and their corresponding biological activities highlighted the potential health benefits associated with the consumption of Podolica milk. Among the proteins identified, in fact, Secretin plays an important role in regulating duodenal pH, food intake, and water homeostasis, indicating its relevance to digestive health. The presence of such proteins highlights the complexity of the milk’s proteome, but also suggests that Podolica milk may contribute to various physiological processes such as homeostasis and food intake. The proteomics/peptidomics approach used allowed the detection of 2213 peptides whose origins may be due to unspecific protein cleavages (multiple peptidases). Among these, 14 bioactive peptides were found, corresponding to 19 hits on MBPDB. As visible in Figure 1, Podolica milk showed a much higher number of bioactive peptides underlining the peculiar nutritional value of this product. More precisely, among five bioactive peptides present in milk from control cows, four were commonly shared with Podolica milk. On the other hand, 10 bioactive peptides were specifically found only in Podolica milk.

The Gene Ontology (GO) analysis obtained with UNIPEPT highlighted the predominant biological processes linked to the detected peptides, including immune response and antimicrobial activity against bacteria that cause infections. This implies that Podolica milk could serve as a functional food, potentially offering protective benefits against some harmful Gram-positive bacteria and supporting the immune system.

*Glucose homeostasis* (Table 3, green)—Two peptides exhibited dipeptidyl peptidase 4 (DPP-IV) inhibitory activity, which is beneficial for glucose metabolism and may help manage diabetes. Specifically, the DPP-IV-inhibitory peptides identified originated from alpha-lactalbumin and are recorded with an IC50 of 13.8 µM for the most powerful one, indicating a strong potential for therapeutic applications [20].

*Anti-hypertensive* (Table 4, blue)—The detection of ACE-inhibitory peptides across multiple casein proteins (alpha-lactalbumin, alpha-S1-casein, and kappa-casein) reinforces the nutritional importance of these milk components in the possible modulation of hypertension and other related health issues. The peptide LDQWLCEKL, corresponding to the amino acid sequence of (115–123) in α-lactalbumin, exhibited a half-maximal inhibitory concentration (IC50) of 131 μM [20]. Furthermore, the detection and identification of peptides such as VGINYWLAHK, FFVAPFPEVFGK, FPEVFGK, PFPEVFGK, YLGY, and HPHPHLSF demonstrates the ACE-inhibition properties originated from alpha-S1-casein and kappa-casein fragments. Those peptides exhibit varying degrees of ACE inhibitory activity, as denoted by their IC50 values, ranging from high potency (e.g., YLGY with IC50 of 9.87 μM) to moderate efficacy (e.g., VGINYWLAHK with IC50 of 327 μM) (Table 4). The peptides derived from alpha-S1-casein, including FFVAPFPEVFGK, which shows multiple IC50 values depending on assay conditions and sources, suggest the potential of casein hydrolysates to produce a variety of bioactive fragments [22]. Such diversity in ACE-inhibitory peptides supports the hypothesis that milk proteins, particularly under controlled enzymatic hydrolysis, can be a source of peptides with significant bioactivity, especially for cardiovascular health applications. These peptides retain bioactivity even post-digestion, a critical factor for their efficacy in vivo as the therapeutic potential of the identified ACE-inhibitory peptides depends on their stability and resistance to gastrointestinal digestion. Peptides like YLGY, which showed an IC50 of 41.86 μM, retain bioactivity after exposure to digestive enzymes [21,23,24,26,27,28,29,30].

*Oxidative stress reduction* (Table 5, orange)—The peptides VGINYWLAHK and YLGY, derived from bovine alpha-lactalbumin and alpha-S1-casein, have antioxidant properties, contributing to oxidative stress reduction, as reported by Chen et al. and Contreras et al. (Table 5) [31]. Antioxidant peptides like these can play a vital role in health maintenance by scavenging free radicals and protecting cells from oxidative damage.

*Calcium metabolism* (Table 6, grey)—The peptides DIGSESTEDQAMEDIK and YKVPQLEIVPNSAEER, have been shown to enhance calcium uptake, a property particularly beneficial for bone health. These peptides, derived from alpha-S1-casein, increase calcium bioavailability by promoting uptake in Caco-2 cell monolayers, as described by Cao et al. [32]. Such multifunctional peptides not only support cardiovascular health but also present potential applications in bone health supplements and functional foods targeting multiple health benefits.

*Antimicrobial activity* (Table 7, pink)—Several detected antimicrobial peptides (AMPs) derived from alpha-S1-casein and showed effects against various bacteria. In particular, the peptides SDIPNPIGSENSEK, VLNENLLR, and YLEQLLR exhibit antimicrobial activity against both Gram-positive and Gram-negative bacteria, as indicated by their minimum inhibitory concentrations (MICs) (Table 7). Specifically, SDIPNPIGSENSEK effectively inhibited *L. innocua* that is an indicator organisms of *L. monocytogenes*, suggesting its potential as a natural antimicrobial against this foodborne pathogen [33]. The peptide VLNENLLR demonstrated a broader antimicrobial spectrum, inhibiting the most important groups of bacteria such as *E. coli, C. sakazakii, L. innocua, L. bulgaricus,* and *S. mutans*, as described by Hayes et al. and Kent et al. in both the clinical and food sectors. Furthermore, YLEQLLR exhibited significant MIC values against *B. subtilis* and two *E. coli*, underscoring its potential utility in controlling some spoilage bacteria and foodborne pathogens [36].

*Miscellaneous activity* (Table 8, white)—This section summarizes the remaining identified peptides with diverse activities. The peptide “YLGYLEQLLR”, from alpha-S1-casein, has been noted for its antianxiety properties. Studied within the framework of infant formula digestion, this peptide appears to exert a calming influence, which may be beneficial in managing stress [37]. The peptide HPHPHLSF, commonly shared between Bos taurus and *Ovis aries* and originating from kappa-casein has a major role in promoting osteoblast differentiation. The osteoanabolic activity of HPHPHLSF thus represents a promising step toward dietary solutions that support skeletal health, which could become integral in aging-related health strategies [38]. The peptide YIPIQYVLSR, from *Bos taurus* kappa-casein, exhibits both immunomodulatory and opioid-like effects. On the one hand, as an agonist for C3a receptors, it may help regulate immune responses, offering benefits for individuals with compromised or overactive immune systems [39]. On the other hand, its opioid-like properties point to potential applications in pain management and mood enhancement [40,41]. This dual functionality positions YIPIQYVLSR as a remarkable natural compound, blending immune support with pain relief and mental well-being effects. Such multifaceted activity highlights how milk peptides may address complex health issues through natural, bioactive compounds.

In summary, this investigation into the bioactive peptides present in Podolica milk not only demonstrates its nutritional potential but also lays the groundwork for future studies aimed at harnessing these bioactive compounds for health benefits. Given the increasing consumer interest in functional foods, the findings of this study could contribute to the valorization of Podolica milk, supporting rural economies and promoting sustainable farming practices.

## 5. Conclusions

With this study we detected bioactive peptides in Podolica milk, demonstrating its potential as a functional food with diverse health-promoting properties. Using a top-down proteomic approach, we identified key proteins, including Secretin, which plays a role in digestive regulation, food intake, and water homeostasis. A total of 14 detected peptides were primarily derived from alpha-lactalbumin, alpha-S1-casein, and kappa-casein, with functions spanning from metabolic regulation to antimicrobial defense.

Two peptides were annotated with dipeptidyl peptidase-IV (DPP-IV)-inhibitory activity, with one from alpha-lactalbumin showing an IC50 of 13.8 µM, indicating potential for glucose metabolism regulation and Type 2 diabetes management. Six peptides exhibited angiotensin-converting enzyme (ACE) inhibitory activity, suggesting cardiovascular benefits. Additionally, two peptides showed antioxidant properties, while two others contributed to calcium uptake enhancement, which is crucial for bone health.

Three antimicrobial peptides, all derived from alpha-S1-casein, exhibited activity against *L. innocua*, *E. coli*, *C. sakazakii*, and *B. subtilis*, reinforcing the potential of Podolica milk as a natural antimicrobial source.

The Gene Ontology (GO) analysis confirmed the predominance of immune and antimicrobial responses, particularly against Gram-positive bacteria, emphasizing the protective role of these peptides. The identification of bioactive peptides with conserved sequences between *Bos taurus* and *Ovis aries* highlights the evolutionary relevance of these molecules in dairy products.

Overall, these findings highlight the potential of Podolica milk as a natural source of health-beneficial bioactive peptides, supporting its role in functional nutrition and therapeutic applications. Additionally, these insights contribute to the valorization of Podolica milk, promoting its role in sustainable dairy farming and rural economic development.

The specific composition of the peptides of Podolica milk concerning the glucose metabolism mainly represent anti-hypertensive and antimicrobial peptides that might act in synergy, amplifying the pharmacodynamic effect. Future developments will be focused on the synthesis of the blend of the peptides responsible for these functions to test and possibly confirm (in vitro and in vivo) the proposed hypothesis.

## Figures and Tables

**Figure 1 foods-14-00877-f001:**
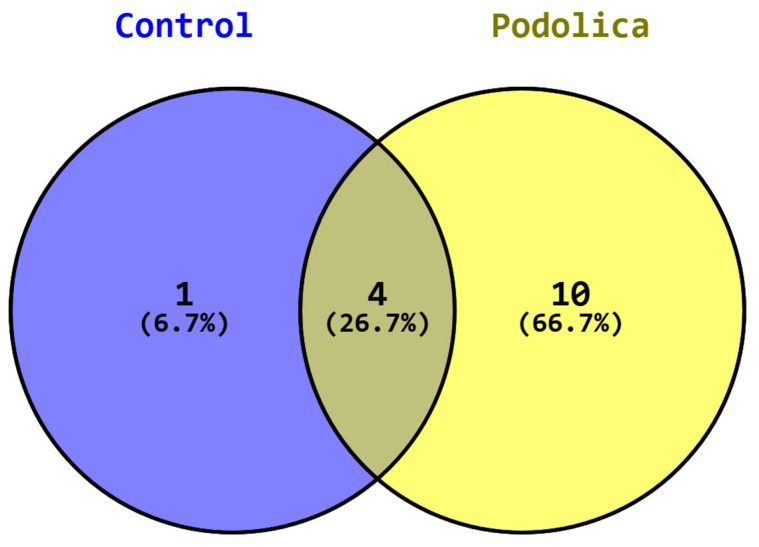
Venn diagram indicating peptide distribution among control and Podolica samples.

**Figure 2 foods-14-00877-f002:**
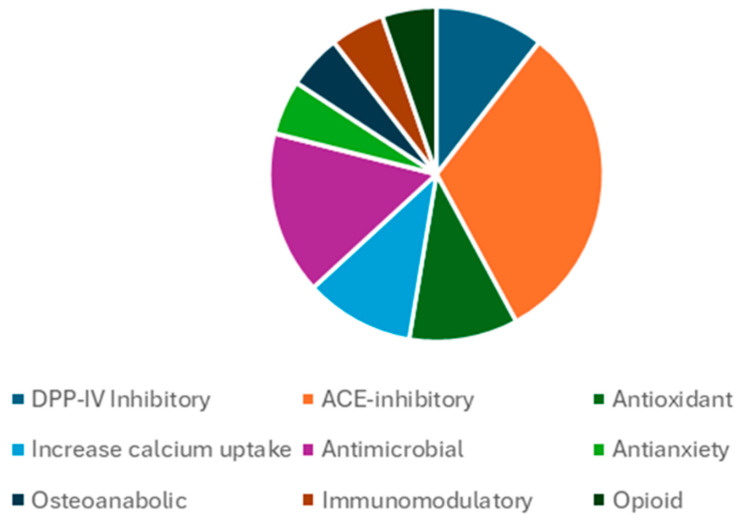
Pie-chart highlighting most represented functions of detected peptides.

**Figure 3 foods-14-00877-f003:**
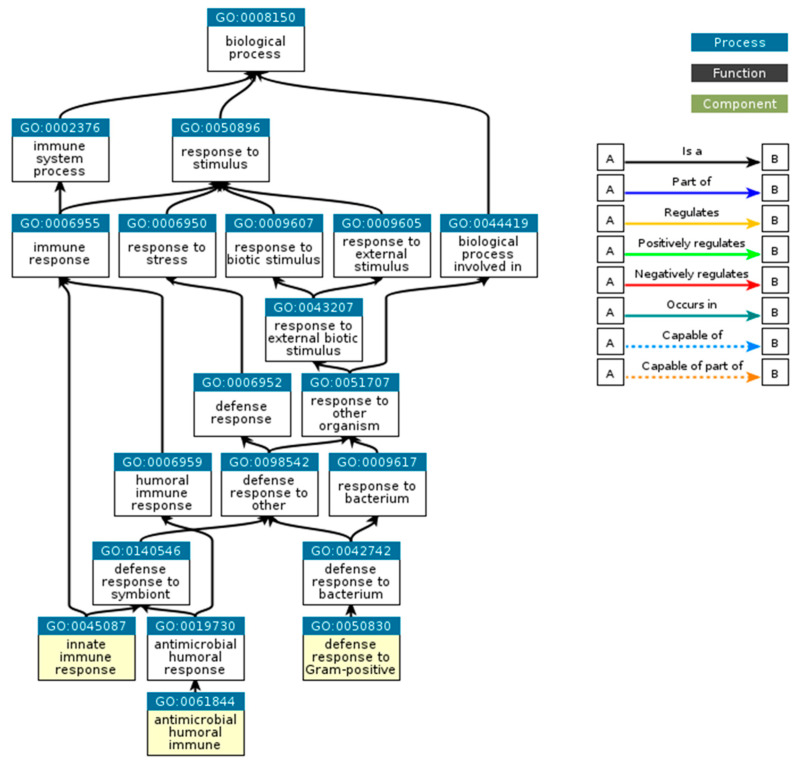
UNIPEPT GO (https://www.ebi.ac.uk/QuickGO/, accessed on 29 July 2024) analysis highlighting biological processes. The most specific GO-terms are represented with the yellow background.

**Figure 4 foods-14-00877-f004:**
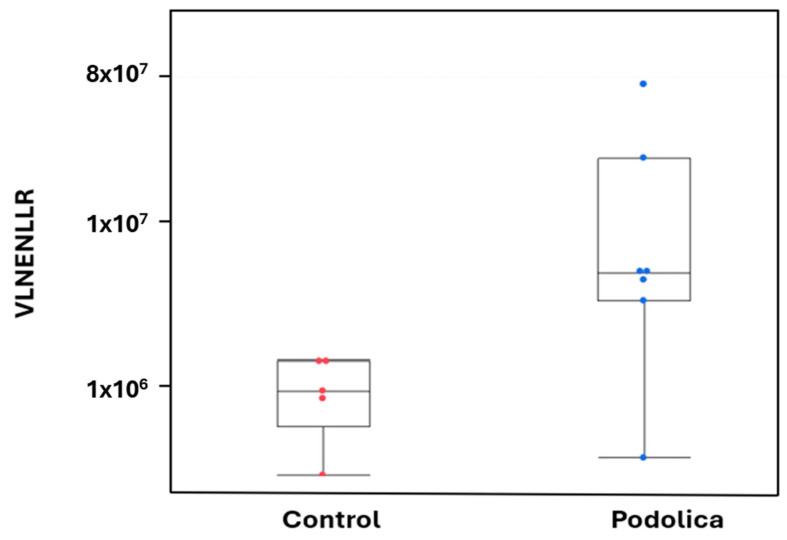
Peptide with antimicrobial function most represented in Podolica cow breeds.

**Table 1 foods-14-00877-t001:** Proteins identified using entire *Bos taurus* database with no cleavages allowed.

Sequence	Mass	Protein	AA Residues	Protein Name	Charges	Score
HSDGTFTSELSRLRDSARLQRLLQGLV	3054.632	P63296	27	Secretin	3	15.469
VELQNRLSDESQ	1416.685	P60316	12	Protein Hook homolog 3	1; 2; 3	22.339

**Table 2 foods-14-00877-t002:** Bioactive peptides present in Podolica milk precipitate. Number after colon indicates number of peptides.

	Hits	Functions	Protein IDs
Grouped Results:		DPP-IV-inhibitory: 2 ACE-inhibitory: 6 Antioxidant: 2 Increase calcium uptake: 2 Antimicrobial: 3 Antianxiety: 1 Osteoanabolic: 1 Immunomodulatory: 1 Opioid: 1	P00711: 4 P02662: 11 P02669: 2 P02668: 2
Total Counts:	19	9	4

**Table 3 foods-14-00877-t003:** Identified peptides with demonstrated DPP-IV-inhibitory function.

Peptide	Protein ID	Protein Description	Intervals	Function	IC50 (μM)
** *LDQWLCEKL* **	P00711	Alpha-lactalbumin	134–142	DPP-IV-Inhibitory [20]	59.6
** *VGINYWLAHK* **	P00711	Alpha-lactalbumin	118–127	DPP-IV-Inhibitory [20]	13.8

**Table 4 foods-14-00877-t004:** Identified peptides with ACE-inhibitory function.

Peptide	Protein ID	Protein Description	Intervals	Function	IC50 (μM)
** *VGINYWLAHK* **	P00711	Alpha-lactalbumin	118–127	ACE-inhibitory [21]	327
** *FFVAPFPEVFGK* **	P02662	Alpha-S1-casein	38–49	ACE-inhibitory	(1) 52.0 [22]
					(2) 77.0 [23]
					(3) 18.0 [24]
** *FPEVFGK* **	P02662	Alpha-S1-casein	43–49	ACE-inhibitory	140 [25]
** *PFPEVFGK* **	P02662	Alpha-S1-casein	42–49	ACE-inhibitory	108 [26]
** *YLGY* **	P02662	Alpha-S1-casein	106–109	ACE-inhibitory	(1) 41.86 [27]
					(2) 9.87 [28]
** *HPHPHLSF* **	P02669	Kappa-casein	119–126	ACE-inhibitory	(1) 28.9 [29]
					(2) 28.9 [30]

**Table 5 foods-14-00877-t005:** Table resuming peptides with antioxidant function.

Peptide	Protein ID	Protein Description	Intervals	Function
** *VGINYWLAHK* **	P00711	Alpha-lactalbumin	118–127	Antioxidant [31]
** *YLGY* **	P02662	Alpha-S1-casein	106–109	Antioxidant [27,28]

**Table 6 foods-14-00877-t006:** List of peptides involved in increased calcium uptake.

Peptide	Protein ID	Protein Description	Intervals	Function	PTM
** *DIGSESTEDQAMEDIK* **	P02662	Alpha-S1-casein	58–73	Increase calcium uptake [32]	2 phosphorylations
** *YKVPQLEIVPNSAEER* **	P02662	Alpha-S1-casein	119–134	Increase calcium uptake [32]	1 phosphorylation

**Table 7 foods-14-00877-t007:** Peptides with antimicrobial function.

Peptide	Protein ID	Protein Description	Intervals	Function	Inhibition Type	Inhibited Microorganisms
** *SDIPNPIGSENSEK* **	P02662	Alpha-S1-casein	195–208	Antimicrobial [33]	MIC	*L. innocua*
** *VLNENLLR* **	P02662	Alpha-S1-casein	30–37	Antimicrobial	(1) MIC,	(1) *E. coli C. sakazakii L. innocua L. bulgaricus S. mutans* [33]
					(2) MIC,	(2) *C. sakazakii* [34]
					(3) MIC	(3) *C. sakazakii C. muytjensii* [35]
** *YLEQLLR* **	P02662	Alpha-S1-casein	109–115	Antimicrobial	MIC	*B. subtilis*—53.6 *E. coli* NEB 5α—241 *E. coli* ATCC 25,922—40.2 [36]

**Table 8 foods-14-00877-t008:** Peptides with miscellaneous bioactivity.

Peptide	Protein ID	Protein Description	Intervals	Function
**YLGYLEQLLR**	P02662	Alpha-S1-casein	106–115	Antianxiety [37]
**HPHPHLSF**	P02669	Kappa-casein	119–126	Osteoanabolic [38]
**YIPIQYVLSR**	P02668	Kappa-casein	46–55	Immunomodulatory [39]
**YIPIQYVLSR**	P02668	Kappa-casein	46–55	Opioid [40,41]

## Data Availability

The original contributions presented in this study are included in the article/Appendix A. Further inquiries can be directed to the corresponding author.

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
