# Peer review of "Detection of Bioactive Peptides’ Signature in Podolica Cow’s Milk"

_foods, 2025, doi:10.3390/foods14050877_

Round 1
Reviewer 1 Report (Previous Reviewer 1)
Comments and Suggestions for Authors
Discussion should be integrated to one paragraph.
Author Response
Discussion should be integrated to one paragraph.
Response: Many thanks for this comment. The discussion definitely needed to be revised. We merged into single paragraphs the different sections and provided sub-headings to provide shape and order. Thanks for reading thoroughly our manuscript and for the precious suggestions.
Reviewer 2 Report (Previous Reviewer 2)
Comments and Suggestions for Authors
The article presents a comprehensive study on the identification and characterization of bioactive peptides in Podolica cow’s milk, a traditional breed from Southern Italy. The research is well-structured, employing advanced proteomic and peptidomic techniques to uncover the functional properties of these peptides. The findings are significant, as they highlight the potential health benefits of Podolica milk, including its role in managing diabetes, cardiovascular health, and antimicrobial activity. The study also aligns with sustainable agricultural practices, which is a strong point. Suggestions for Revision are as following:
- Clarity and Organization:
Introduction: The introduction could be more concise. Some sections, particularly the background on Podolica cattle and milk, could be streamlined to focus more on the bioactive peptides and their significance. (Suggested reduction: ~10 lines)
*Results Section: The results are detailed but could be better organized. Consider using subheadings to separate different types of bioactive peptides (e.g., DPP-IV inhibitors, ACE inhibitors, antimicrobial peptides) for easier readability. (Suggested reorganization: ~5 lines)
- Discussion:
- The discussion is thorough but somewhat repetitive. Some points, such as the potential of Podolica milk in managing diabetes and hypertension, are reiterated multiple times. Condensing these sections would improve the flow.
- The discussion on the evolutionary conservation of bioactive peptides across species (e.g., Bos taurus and Ovis aries) is interesting but could be more concise.
- Figures and Tables:
Figure 1 (Venn Diagram): The Venn diagram is useful but could be better explained in the text. Consider adding a brief description of what the overlapping and unique peptides signify in terms of functional properties.
Tables: The tables are informative but could be more visually appealing. Consider using color coding or shading to differentiate between different types of bioactive peptides (e.g., DPP-IV inhibitors, ACE inhibitors).
- Conclusion:
The conclusion is well-written but could be more forward-looking. Consider adding a brief paragraph on the potential for future research, such as in vivo studies or clinical trials to validate the health benefits of these peptides.
- Supplementary Materials:
- The supplementary materials (e.g., Figure S1) should be briefly described in the main text to guide readers on where to find additional information.
- Language and Style:
Some sentences are overly complex and could be simplified for better readability. For example, the sentence "The presence of such proteins not only highlights the complexity of the milk’s proteome but also suggests that Podolica milk may contribute to various physiological processes such as homeostasis and food intake" could be broken down into simpler sentences.
Author Response
The article presents a comprehensive study on the identification and characterization of bioactive peptides in Podolica cow’s milk, a traditional breed from Southern Italy. The research is well-structured, employing advanced proteomic and peptidomic techniques to uncover the functional properties of these peptides. The findings are significant, as they highlight the potential health benefits of Podolica milk, including its role in managing diabetes, cardiovascular health, and antimicrobial activity. The study also aligns with sustainable agricultural practices, which is a strong point.
Suggestions for Revision are as following:
- Clarity and Organization:
Introduction: The introduction could be more concise. Some sections, particularly the background on Podolica cattle and milk, could be streamlined to focus more on the bioactive peptides and their significance. (Suggested reduction: ~10 lines)
Response: The introduction was amended accordingly. Many thanks to the referee for this valuable comment as it is better to focus on traceability and bioactivity of the peptides rather than on the story of these animals.
*Results Section: The results are detailed but could be better organized. Consider using subheadings to separate different types of bioactive peptides (e.g., DPP-IV inhibitors, ACE inhibitors, antimicrobial peptides) for easier readability. (Suggested reorganization: ~5 lines)
Response: On behalf of all the authors, thanks for this comment. Little subheadings are now indicated in the results section for each function.
- Discussion:
- The discussion is thorough but somewhat repetitive. Some points, such as the potential of Podolica milk in managing diabetes and hypertension, are reiterated multiple times. Condensing these sections would improve the flow.
Response: The discussion and, particularly, the mentioned section were amended accordingly. Thank you for the thorough reading and for the suggestion.
- The discussion on the evolutionary conservation of bioactive peptides across species (e.g., Bos taurus and Ovis aries) is interesting but could be more concise.
Response: All authors agreed, and the section was amended accordingly.
- Figures and Tables:
Figure 1 (Venn Diagram): The Venn diagram is useful but could be better explained in the text. Consider adding a brief description of what the overlapping and unique peptides signify in terms of functional properties.
Response: Many thanks for this really spot-on comment as the description of the Venn diagram was very poor. Now it description has been improved in lines 216-221 along with some brief information about the bioactivity of the unique and commonly-shared peptides.
Tables: The tables are informative but could be more visually appealing. Consider using color coding or shading to differentiate between different types of bioactive peptides (e.g., DPP-IV inhibitors, ACE inhibitors).
Response: We think that it is a great idea. All the tables have been amended.
- Conclusion:
The conclusion is well-written but could be more forward-looking. Consider adding a brief paragraph on the potential for future research, such as in vivo studies or clinical trials to validate the health benefits of these peptides.
Response: Thanks for the comment. The peptides’ function was already demonstrated for the single ones. However, our idea for the further development of this research, would be to study the eventual synergistic effect of the peptides originating form the proteins of this milk. A brief paragraph was added right at the end of the conclusion section.
- Supplementary Materials:
- The supplementary materials (e.g., Figure S1) should be briefly described in the main text to guide readers on where to find additional information.
Response: Thanks for this important comment that helps to add scientific value to our research. SF1 has now been commented in lines 198-201 highlighting that, the overall composition of high abundant proteins of Podolica and control milk is pretty much the same. Now all was described in the text.
- Language and Style:
Some sentences are overly complex and could be simplified for better readability. For example, the sentence "The presence of such proteins not only highlights the complexity of the milk’s proteome but also suggests that Podolica milk may contribute to various physiological processes such as homeostasis and food intake" could be broken down into simpler sentences.
Response: All authors agreed. We read again the entire manuscript to try to minimize these frequent complex sentences construction.
Dear referee 2, I would like to personally thank you for your constructive feedback that made our manuscript better.
Reviewer 3 Report (Previous Reviewer 3)
Comments and Suggestions for Authors
The controversial tables are improved. I had no other serious charges against the manuscript.
Author Response
The controversial tables are improved. I had no other serious charges against the manuscript.
Response: Thanks for your valuable feedback and response.
This manuscript is a resubmission of an earlier submission. The following is a list of the peer review reports and author responses from that submission.
Round 1
Reviewer 1 Report
Comments and Suggestions for Authors
"Detection of bioactive peptides’ signature in Podolica cow’s milk" was prepared to pave the way for exploring Podolica milk in developing health-promoting dairy products such as cheeses and supplements. However, there are many parts needing to be imrpoved. Some comments are listed as follow.
1. Introduction should be improved. It is lack of the recent research on the
bioactive peptides. Lots of content is derived from one article.
2. In Table 3, 4, and 5, the cited articles should be listed in the references.
3. Conclusion should be rewritten to show the main results.
Reviewer 2 Report
Comments and Suggestions for Authors
This article is about the bioactive peptides’ signature in Podolica cow’s milk. The topic is novel, the manuscript is well written. However, the design of the study is not very scientific. At lest 3 samples should be collected at each time, but only I was collected in the study.
The proteins identified in your study should also be exhibited in results.
You shold also give the total number of peptides in different goups.
Sometimes Venn figure of peptide composition is better than table.
Line 179: 8 samples were mentioned in methods, why only 7 here?
Line 184: Didn’t you identify whey protein in your study?
You should also compare the difference in peptides between different farms or different time.
Line 300: XXXX?
Reviewer 3 Report
Comments and Suggestions for Authors
I have recommended rejection due to ethical concerns i.e. inappropriate use of tables form the MBPDB database.
My ethical concerns apply copying tables from Milk Bioactive Peptide Database (MBPDB). Search results in the above database are displayed as tables. Authors simply copied content of tables from MBPDB. Three columns, entitled: "Abstract", Search type" and "Scpring matrix" are removed. All other are copied "as they were". Example of MBPDB table layout using peptide with sequence LDQWLCEKL (first peptide in the Table 3) is presented in the attached file Screenshots.pdf. I have added also screenshot of database homepage with peptide sequence used as a query. Tables 3-8 (5 pages from 18 in the manuscript) are copied from the MBPDB database.
Authors have stated in Materials and methods section, thet they use MBPDB database, thus I do not consider presentation of results as plagiarism (It may be consulted with lawyer). Of course data from MBPDB is publicly available and can be used in publications, but table layout change is obligatory. Sinple copying is unacceptable.
Some columns in tbales 3-8 are not necessary. Column entitled "Species" is not necessary due to fact that Authors have stated in title, that they used cow's milk. Placemnt in the column ewe latin name is confusing. Ovine milk was not substrate for experiment. To be honest, Authors have provided correct explanation in the text, but is not rationale to presence this column in the tables. Column entitled "Function" only repeats information from table captions. There is no rationale for columns entitled "Title", "Authors" and "DOI". Tables should contain only article number. Full bibliographic data should be provided in the "References" section. Authors could state, that protein ID numbers (column "Protein ID") are taken from the UniProt database.
Generally bioinformatics tools with except of MBPDB. Authors should cite not only website, but also articel describing database. For instance the latest article describin UniProt has been published few days ago in the Nucleic Acids Research journal (Database issue). Publications describing UniPept are summarized at the program website. References for MaxQuant program and CARD database are also available.
Taking above into account I do not recommend the manuscript for publication.
I'm sorry.
